# Neutrophil-to-Lymphocyte Ratio, Neutrophil-to-Monocyte Ratio, Platelet-to-Lymphocyte Ratio, and Systemic Immune-Inflammation Index in Psoriasis Patients: Response to Treatment with Biological Drugs

**DOI:** 10.3390/jcm12175452

**Published:** 2023-08-22

**Authors:** Hulya Albayrak

**Affiliations:** Dermatology Department, Faculty of Medicine, Namık Kemal University, Tekirdağ 59030, Turkey; halbayrak@nku.edu.tr

**Keywords:** psoriasis, neutrophil-ymphocyte ratio, platelet-lymphocyte ratio, neutrophil-monocyte ratio, systemic immune-inflammation index

## Abstract

Background: Psoriasis is a chronic immune-mediated skin disease in which systemic inflammation plays an important role in its pathogenesis. In recent years, the neutrophil-to-lymphocyte ratio (NLR), neutrophil-to-monocyte ratio (NMR), platelet-to-lymphocyte ratio (PLR), and systemic immune-inflammation index (SII) were shown to be important indicators of inflammation. This study aimed to investigate the NLR, NMR, PLR, and SII levels in psoriasis patients treated with biological agents. Method: Clinical and biochemical data of 209 patients who received systemic therapy for psoriasis were obtained by retrospectively reviewing their medical records. The NLR, NMR, PLR, and SII values were calculated from the hemogram values of the patients. Results: In the third month of follow-up, the mean CRP, NLR, NMR, PLR, and SII values were significantly decreased compared with the baseline values. The SII values showed strong positive correlations with the NLR, NMR, and PLR. Adalimumab, etanercept, and infliximab, which are TNF-α blockers, were observed to be more effective on the PLR and NLR, and especially the NMR. Conclusions: The NLR, NMR, PLR, and SII, which are data derived from routine blood tests, can be used in the monitoring of the treatment of psoriasis, especially with TNF-α blockers.

## 1. Introduction

Psoriasis is a chronic systemic inflammatory disease that can affect the skin and/or joints, affecting 2–3% of the world’s population [1]. As in many inflammatory diseases, it is important to determine the severity of the disease for safe and effective treatment. As there is no single tool that can fully assess the severity of psoriasis, the assessment of the disease becomes more complex [2]. The psoriasis area severity index (PASI), which evaluates the degree of induration, erythema, and desquamation in the affected body parts, has been one of the most commonly used scales since 1978 [3]. In cases where the PASI cannot be performed, the body surface area (BSA) distribution percentage is another simple scale used. In addition, parameters used in routine blood tests, such as CRP, cytokines, and adhesion molecules in psoriasis patients have also been used to evaluate disease activity [4,5,6]. However, new parameters are required due to the lack of objective evaluation criteria, insufficient evaluation of chronic inflammation-causing disease, and differences between clinicians.

Due to the role of inflammation in the pathogenesis of psoriasis and the increase in our knowledge about cytokines, interleukins, and serum autoantibodies over time, better markers for the diagnosis and severity of psoriasis are available. Therefore, the identification of widely used and low-cost biomarkers of systemic inflammation that play a role in the pathogenesis of psoriasis, in addition to the clinical data of the patients, may be useful for the evaluation of psoriasis patients. Recently, many indexes related to systemic inflammation from routine complete blood count (CBC) tests have been used because of their ability to predict outcomes in pathological conditions [7]. The platelet-to-lymphocyte ratio (PLR), neutrophil-to-lymphocyte ratio (NLR), and neutrophil-to-monocyte ratio (NMR), which can be easily derived as a part of a routine CBC, are widely used in chronic inflammatory diseases as simple markers of systemic inflammatory response [5,8]. Studies have investigated the use of the PLR, NLR, and NMR as markers of systemic inflammation in rheumatoid arthritis, ulcerative colitis, acute coronary syndrome, diabetes mellitus, end-stage renal disease, tuberculosis, familial Mediterranean fever, and cirrhosis [8,9,10,11,12,13]. In addition, studies used the PLR and NLR to determine the prognosis in cancer patients [14]. There are studies suggesting that the PLR and NLR are increased in psoriasis patients compared with controls, and therefore, they can be easy and inexpensive markers for psoriasis patients [15,16,17]. Similar to the PLR and NLR, studies show that the systemic immune-inflammation index (SII), which is another marker obtained from a routine CBC, can be used as an effective prognostic factor in diseases where chronic inflammation is causative [18]. Parmana et al.’s [19] findings of the SII’s significance in off-pump coronary artery bypass surgery, underscores the importance of biomarkers in medical prognostication. Moldovan et al. [20] illustrated the correlation between the NLR, PLR, MLR, and SII ratios with the invasiveness of surgical procedures in elderly hip fracture patients, identifying noteworthy associations and establishing that elevated postoperative SII levels serve as a dependable indicator of surgical trauma.

The immune axis of tumor necrosis factor (TNF)-α/interleukin (IL)-23/IL-17 is involved in the pathogenesis of psoriasis. Therefore, biologics targeting these cytokines have been used in the treatment of psoriasis in recent years [21]. Although TNF-α inhibitors show high efficacy against rash, their effects on the PLR, NMR, NLR, and SII, which are the markers of the inflammatory reaction, have not been adequately studied. In other words, the answer to whether the PLR, NLR, NMR, and SII can be biomarkers for the efficacy of TNF-α inhibitors has not been fully elucidated. Therefore, the aim of this retrospective study was to investigate the effects of the TNF-α blockers adalimumab, etanercept, and infliximab; the IL-17A antagonists ustekinumab, secukinumab, and ixekizumab; and acitretin and methotrexate, which are used in the treatment of psoriasis, on the PLR, NLR, NMR, and SII in psoriasis patients, and to seek answers to the question of whether these parameters can be biomarkers of treatment efficacy for these drugs.

## 2. Material Methods

### 2.1. Study Design and Patients

This retrospective study was carried out on patients who were followed up and treated with the diagnosis of psoriasis between 2015–2021 in the Dermatology Clinic of the Namık Kemal University Faculty of Medicine. The study protocol was approved by the Namık Kemal University Ethical Committee (identifiers: clinical ethical approval no. 2021.39.02.02). Patients aged 18 years and older with a diagnosis of psoriasis and who had received systemic treatment for psoriasis were included in the study. Patients under 18 years of age; patients with a malignant tumor, active infection, or any systemic inflammatory disease; and patients whose hematological data could not be obtained were excluded from the study.

### 2.2. Sample Collection

All sample collection was conducted in accordance with ethical guidelines and regulatory standards. In the context of this retrospective study, the majority of the patients’ blood samples were procured during the morning hours. Subsequently, the derived outcomes were sourced from the hospital’s comprehensive registration system.

### 2.3. Complete Blood Count (CBC) Determinations

The complete blood count (CBC) determinations were performed using a state-of-the-art hematology analyzer, namely, the Mindray bc 6000. This analyzer has been well-established for its precision and accuracy in providing comprehensive blood cell counts and related parameters. The CBC measurements encompassed a range of parameters, including but not limited to red blood cell count (RBC), white blood cell count (WBC), hemoglobin concentration (HGB), hematocrit (HCT), mean corpuscular volume (MCV), mean corpuscular hemoglobin (MCH), mean corpuscular hemoglobin concentration (MCHC), neutrophil count, lymphocyte count, eosinophil count, basophil count, monocytes count, platelet count (PLT), mean platelet volume (MPV), platelet distribution width (PDW), C-reactive protein (CRP), and erythrocyte sedimentation rate (ESR). The analytical methodology, grounded in impedance principles, facilitated the extraction of the aforementioned measurements.

### 2.4. Definition of Systemic Inflammatory Markers

The NLR was obtained by dividing the neutrophil count by the lymphocyte count, the NMR was obtained by dividing the neutrophil count by the monocyte count, and the PLR was obtained by dividing the platelet count by the lymphocyte count. The SII was calculated with the formula neutrophil (N) × platelet (P)/lymphocyte (L) (SII = N × P/L ratio) and recorded.

NLR—neutrophil count/lymphocyte count;NMR—neutrophil count/ monocyte count; PLR—platelet count/lymphocyte count;SII—neutrophile count × (platelet count/lymphocyte count ratio).

Age, gender, disease duration, PASI values, hematological values, and treatments applied to the patients (methotrexate, adalimumab, acitretin, ustekinumab, etanercept, secukinumab, ixekizumab, and infliximab) were recorded using hospital information and compared according to the treatment groups.

### 2.5. Statistical Analysis

The statistical analysis was performed using SPSS 20 statistical software. The Shapiro–Wilk test was used to evaluate the suitability of the measured data to the normal distribution. The mean, standard deviation, median, minimum, and maximum values of continuous variables, and n and percentage values of categorical variables were given. The one-way ANOVA test was used for the analysis of normally distributed data in the comparison between groups, and the Kruskal–Wallis test was used for data that did not show a normal distribution. The Friedman test was used to compare repeated measurements. If there was a difference between the measurements, the Wilcoxon test was used for pairwise comparison. Spearman’s test was performed for the correlation of hematological parameters. For all statistics, *p* < 0.05 was determined to be significant.

## 3. Results

The demographic characteristics of all participants included in the study are shown in Table 1. The mean age of the participants was 42.40 ± 13.01 years, the mean PASI value was 11.76 ± 6.42, and the mean time post-onset was 13.73 ± 10.27 years. Of the 209 participants, 129 were male and 80 were female. A total of 64 patients were using methotrexate, 43 patients were using acitretin, 40 patients were using ustekinumab, and 26 patients were using adalimumab.

The change in hematological values of the participants over time with treatment is shown in Table 2. In the third month of follow-up, the mean RBC, neutrophil, platelet, CRP, ESR, NLR, NMR, PLR, and SII values decreased significantly compared with the baseline values, while the mean MCV, RDW, lymphocyte, monocyte, MPW, and PDW values increased significantly compared with the baseline values. In the sixth month of follow-up, the mean RBC, HCT, neutrophil, NLR, and PLR values increased significantly compared with the third month. However, although the NMR and PLR increased, they were still significantly lower than the baseline. While the mean MCH, WBC, monocyte, PDW, CRP, ESR, and NMR values decreased significantly at the sixth month of follow-up compared with the third month, the mean MCH, CRP, ESR and NMR values at the sixth month were still significantly different from the baseline values.

Adalimumab, acitretin, etanercept, and infliximab significantly decreased the NMR at the third month of treatment compared with the baseline values, while all biologics used in the treatment significantly decreased the NMR at the sixth month of treatment compared with the baseline values (Table 3). Adalimumab, acitretin, etanercept, and infliximab significantly decreased the PLR at the third month of treatment compared with the baseline values, while infliximab, ustekinumab, and adalimumab significantly decreased the PLR at the sixth month of treatment compared with the baseline values. Adalimumab and infliximab significantly decreased the NLR at the third month of treatment compared with the baseline values, while only adalimumab significantly decreased the NLR at the sixth month of treatment compared with the baseline values. Adalimumab treatment alone decreased the SII values at both the third and sixth months of treatment compared with the baseline values (Table 3).

There was no difference between the treatment agents in terms of NLR-NLR3, NLR-NLR6, NMR-NMR3, NMR-NMR6, PLR-PLR3, and PLR-PLR6 changes (Table 4).

Correlations of the NLR, NMR, and PLR with other parameters are given in Table 5. The NLR had weak positive correlations with the PASI and CRP and strong positive correlations with the NMR and PLR. The SII showed weak positive correlations with the PASI and CRP, and strong positive correlations with the PLR, NMR, and NLR. No significant correlations were observed in terms of the other parameters.

## 4. Discussion

In this study, the effects of treatments applied to patients with psoriasis on hematological parameters were investigated. In the third month of follow-up, the mean CRP, NLR, NMR, PLR, and SII values were significantly decreased compared with the baseline values. The SII values showed strong positive correlations with the PLR, NMR, and NLR. Adalimumab, etanercept, and infliximab, which are TNF-α blockers, were observed to be more effective on the PLR, NLR, and especially the NMR.

Unfortunately, there is no laboratory marker to evaluate the disease activity in psoriasis. Therefore, there is a need to identify better markers for the assessment of psoriasis severity and treatment outcomes. In recent studies, the NLR, MLR, and PLR, which can be easily calculated from the neutrophil, monocyte, lymphocyte, and platelet counts, have been defined as markers of inflammation [22]. The NLR, MLR, and PLR, which have been shown to be biomarkers in many diseases associated with systemic inflammation, have also been shown to be associated with psoriasis and the severity of the disease. In controlled studies, it was reported that the NLR is significantly higher in psoriasis patients than in controls, and there is a correlation between the NLR and PASI [3,15,16,17,23]. In a meta-analysis including 1067 psoriasis patients and 799 healthy controls, it was reported that the NLR was higher in psoriasis patients than in healthy controls, and the NLR values were higher in patients with PASI > 10 than in patients with PASI < 10. This meta-analysis also found no correlation between the NLR values and PASI scores [7]. In the same meta-analysis, it was documented that the NLR was correlated with the presence, not severity, of the disease in psoriasis patients. Wang et al., in their study on Chinese psoriasis patients, determined that the NLR was significantly higher in psoriasis patients than in healthy controls. However, they observed that there was no correlation between the NLR and PASI score [24]. The inconsistency in the results of the studies was due to the duration of the disease, the number of cases, the different types of studies, and racial and ethnic differences.

In addition to the NLR, platelets, which are an easily measurable parameter, were found to increase during infectious and inflammatory diseases, including psoriasis [25]. The PLR, which is another value derived from the platelet count, is another suggested indicator of systemic inflammation [26]. Studies showed that the PLR values in psoriasis patients are higher than in healthy controls [3,15]. According to a meta-analysis, the PLR values in psoriasis patients were significantly higher than in healthy controls [7]. In another study that evaluated psoriasis subtypes, it was reported that both the peripheral blood platelet counts and PLR were significantly higher in all four subtypes of psoriasis compared with control subjects [24]. Yurtdaş and colleagues [17] documented that the PLR is high in psoriasis patients and that the PLR correlates with PASI. There were also studies that evaluated the MPV as another biomarker of platelet activation. There were studies that reported that the MPV is higher in psoriatic patients than in healthy controls [27,28,29,30]. They also determined that the MPV showed a positive correlation with the PASI [27,28,30]. However, there were studies that reported that the MPV does not show a significant change in psoriasis patients [31,32] and other studies showed that MPV is lower in patients with psoriasis [33].

It was demonstrated in previous studies that the NLR, PLR, and MLR are increased in psoriasis patients compared with controls and that they can be a biomarker that can be easily used in psoriasis patients. Moreover, there are studies that investigated how the NLR, MLR, and PLR are affected due to treatments in psoriasis patients. In the study by Asahina et al. [33], it was determined that the NLR decreased in psoriasis patients after twelve months of treatment with ustekinumab, infliximab, and adalimumab, and there was no difference in terms of the treatments applied. Moreover, they suggested that the NLR could be used to examine the efficacy of treatment in psoriasis patients. In the study by An et al. [34], it was reported that the NLR decreased after six months of treatment, regardless of the biological agent applied. Aktas Karabay and colleagues [23] observed that PASI scores and NLR values decreased after treatment with narrowband methotrexate, acitretin, ultraviolet B, cyclosporine, ustekinumab, adalimumab, and etanercept in psoriasis patients. Moreover, the reductions in the PASI and NLR showed a positive correlation. In the study of Hagino et al. [35], it was documented that the PASI, NLR, MLR, PLR, and CRP values at 12 and 52 weeks of treatment with infliximab, adalimumab, and certolizumab pegol decreased significantly compared with the baseline. Moreover, it was observed that the NLR, MLR, PLR, and CRP values were mostly correlated with each other before and after treatment with TNF-α inhibitors. In the study by Nobari et al. [36], it was determined that the NLR and PLR decreased significantly after one year of treatment with TNF-α inhibitors. Andersen et al. [37] reported that patients with psoriasis who responded to treatment with TNF-α inhibitors had lower baseline NLR values.

In this study, the mean neutrophil, platelet, CRP, ESR, NLR, NMR, and PLR values were significantly decreased compared with the baseline values in the third month of treatment, regardless of the type of biological agent applied. Although the values increased again at the sixth month of treatment, the NMR and PLR values were still significantly different from the baseline values. The change in parameter values at the sixth month of treatment may have been because the patients did not use their medications regularly. Among the biological agents used in the treatment, the TNF-α blockers adalimumab, etanercept, and infliximab were especially found to be more effective on the NLR, NMR, and PLR. In the sixth month of treatment, the efficacies of all biological agents, especially on the NMR, were more pronounced. No difference was observed in terms of changes in the NLR, PLR, and NMR caused by biological agents. The results of this study especially confirmed the results of previous studies that demonstrated the effectiveness of TNF-α blockers. Again, when the correlation of the NLR, PLR, and NMR with other parameters was examined in this study, it was determined that the NLR was positively correlated with the PASI, CRP, NMR, and PLR, in line with previous studies.

The SII, which is a novel inflammation-based biomarker derived from neutrophils, platelets, and lymphocytes, was recently identified. It was suggested that the SII, like the other hematological inflammatory indices NLR, PLR, and NMR, could be a simple and inexpensive tool to predict the progression of many diseases [38,39,40,41]. In a study conducted with psoriasis patients, it was reported that the SII was significantly higher in these patients than in the controls [42]. Moreover, it was determined that the SII values increased with the increase in the severity of psoriasis. In the same study, it was also observed that there was a positive correlation between the SII and PASI [42]. In a recent study by Liu et al. [43], they determined that the PLR and SII values in patients with psoriatic arthritis were higher than in those with plaque psoriasis. In this study, it was determined that the SII decreased after treatment. While the numerical decrease was observed in other biological agents in the SII after treatment, a significant decrease was determined in adalimumab. In addition, the SII exhibited weak positive correlations with the PASI and CRP, and strong positive correlations with the NLR, PLR, and NMR.

This study revealed a link between the SII levels of psoriasis patients and the treatment they received. This was the strength of this study.

Limitations of this study included the retrospective design, which may introduce inherent biases and incomplete data collection. This study focused on patients treated with biological agents, potentially limiting the generalizability of the findings to other treatment modalities. The study’s duration and the specific time frame for follow-up measurements might affect the comprehensiveness of the observed treatment effects. Additionally, the study primarily relied on clinical and biochemical data without accounting for potential confounding factors or coexisting conditions that could influence the results. The impact of patient characteristics, such as age, comorbidities, and disease severity, on the studied parameters might not have been fully addressed. Lastly, the study’s sample size of 209 patients could affect the statistical power and precision of the results, potentially limiting the ability to detect subtle differences or associations. Since the study was retrospective, it was not known whether the patients used their medications regularly. Therefore, prospective studies with larger numbers of patients are needed.

## 5. Conclusions

This retrospective study aimed to determine a new biomarker from routine blood values for the effectiveness of the treatment applied in psoriasis patients. In this study, a decrease was observed in the NMR, NLR, and PLR as a result of treatment, which is consistent with previous studies. In particular, the NLR correlated with other markers. It was determined that the SII, which is a recently identified biomarker, also decreased as a result of treatment and correlated with other markers. The assessment of the Systemic Immune-Inflammation Index (SII) alongside other hematological parameters offers a valuable approach for appraising the severity of psoriasis and gauging treatment efficacy.

## Figures and Tables

**Table 1 jcm-12-05452-t001:** Demographic and clinical data of the participants.

		Psoriasis Vulgaris
Gender	Male	129 (61.7%)
	Female	80 (38.3%)
Nail psoriasis	No	10 (18.5%)
	Yes	44 (81.5%)
Joint involvement	No	8 (20%)
	Yes	32 (80%)
Biological agent	Metotrexate	64 (30.6%)
	Adalimumab	26 (12.4%)
	Asitretin	43 (20.6%)
	Ustekinumab	40 (19.1%)
	Etanercept	13 (6.2%)
	Secukinumab	9 (4.3%)
	Ixekizumab	6 (2.9%)
	Infliximab	8 (3.8%)
Age (Year)		42.40 ± 13.01, 42 (18–69)
Time post-onset (Year)		13.73 ± 10.27, 11.5 (1–52)
PASI		11.76 ± 6.42, 10 (3.2–36)

PASI: Psoriasis Area Severity Index.

**Table 2 jcm-12-05452-t002:** Comparison of the changes in hematological values of the participants over time.

	Pre-Treatment	After Treatment (3rd Month)	After Treatment (6th Month)
RBC (10^6^/μL)	4.75 ± 0.47 4.73 (3.7–6.5)	4.68 ± 0.63 ^a^ 4.65 (1–7)	4.81 ± 1.04 ^b^ 4.70 (3–15)
HGB (g/dL)	14.26 ± 1.53 14.234 (9.00–18.87)	14.24 ± 1.68 14.40 (10–20)	14.21 ± 1.45 14.20 (10–17)
HCT (%)	42.67 ± 4.23 42.80 (30.50–56.50)	42.49 ± 4.77 42.60 (31–60)	43.14 ± 5.67 ^b^ 42.70 (33–88)
MCV (fL)	90.09 ± 6.48 91.00 (60.0–106.0)	90.64 ± 7.03 92.00 (59–106)	90.46 ± 8.63 91.00 (30.104)
MCH (pg)	23.30 ± 12.80 29.80 (0.00–34.20)	30.37 ± 2.52 ^a^ 30.82 (19.00–35.00)	30.19 ± 2.52 ^a,b^ 30.52 (20–35)
MCHC (g/dL)	33.45 ± 1.10 33.45 (29.0–35.83)	33.52 ± 1.05 33.54 (30.00–36.00)	33.05 ± 1.92 ^a,b^ 33.10 (16–36)
RDW (%)	13.87 ± 1.53 13.55 (11.0–19.30)	14.23 ± 2.48 ^a^ 13.60 (12.00–32.00)	13.84 ± 1.95 13.60 (4–21)
WBC (10^3^/μL)	7.75 ± 2.02 7.55 (393–15.31)	8.18 ± 4.70 4.50 (2.00–59.00)	7.72 ± 1.92 ^b^ 7.60 (4–13)
Neutrophil (10^3^/uL)	4.95 ± 4.63 4.38 (2.00–60.40)	4.44 ± 1.53 ^a^ 4.16 (1.00–9.00)	4.96 ± 3.63 ^b^ 4.38 (2–31)
Lymphocyte (10^3^/uL)	2.37 ± 1.10 2.27 (0.96–13.70)	2.44 ± 0.70 ^a^ 2.30 (1.00–5.00)	2.42 ± 1.17 ^a^ 2.30 (1–13)
Monocyte (10^3^/μL)	0.57 ± 0.18 0.55 (0.20–1.32)	0.61 ± 0.20 ^a^ 0.58 (0.00–1.00)	0.59 ± 0.24 ^b^ 0.56 (0–2)
Eosinophil (10^3^/uL)	0.20 ± 0.12 0.17 (0.04–0.85)	0.20 ± 0.10 0.19 (0.00–1.00)	0.21 ± 0.11 0.17 (0–1)
Basophil (10^3^/μL)	0.03 ± 0.02 0.03 (0.00–0.11)	0.03 ± 0.02 0.03 (0.00–0.12)	0.03 ± 0.02 0.03 (0–0.1)
Platelet (10^3^/μL)	265.80 ± 62.84 259.50 (114.0–408.0)	254.86 ± 69.48 ^a^ 246.00 (71.00–442.00)	254.07 ± 66.63 ^a^ 250.00 (8–417)
MPV (fL)	8.79 ± 0.78 8.80 (7.0–11.0)	8.93 ± 0.093 ^a^ 8.80 (6.90–14.50)	8.89 ± 0.84 ^a^ 8.80 (6–12)
PDW (%)	14.86 ± 2.55 15.00 (0.0–22.50)	15.24 ± 2.58 ^a^ 15.00 (2.00–22.00)	15.01 ± 2.61 ^b^ 14.80 (0–22)
CRP (mg/L)	5.34 ± 9.21 2.65 (0.0–75.0)	4.96 ± 6.45 ^a^ 2.24 (0.40–29.70)	4.88 ± 7.82 ^a,b^ 2.49 (0–48)
ESR (mm/hour)	17.52 ± 18.02 12.0 (1.0–107.0)	13.28 ± 12.44 ^a^ 9.00 (1.00–60.00)	13.50 ± 12.06 ^a,b^ 9.70 (1–51)
NLR	2.21 ± 1.54 1.88 (0.44–17.46)	1.91 ± 0.73 ^a^ 1.85 (0.72–4.55)	2.26 ± 1.79 ^b^ 1.88 (0.44–14.25)
NMR	8.83 ± 5.60 8.01 (2.98–64.26)	7.56 ± 2.42 ^a^ 7.34 (2.46–14.93)	2.42 ± 1.17 ^a,b^ 2.30 (0.59–13.40)
PLR	121.78 ± 40.25 117.17 (23.43–276.24)	111.60 ± 42.81 ^a^ 103.33 (46.86–323.46)	114.57 ± 42.56 ^a,b^ 110.31 (12.88–253.10)
SII	594.76 ± 445.61 500.15 (94.28–4992.60)	504.06 ± 281.31 ^a^ 427.42 (111.26–1644.34)	576.96 ± 455.22 ^a^ 469.96 (26.92–3316.60)

CRP: C-reactive protein, ESR: erythrocyte sedimentation rate, HGB: hemoglobin, HTC: hematocrit, MCH: mean erythrocyte hemoglobin, MCHC: mean corpuscular hemoglobin concentration, MCV: mean erythrocyte volume, MPV: mean platelet volume, NLR: neutrophil–lymphocyte ratio, NMR: neutrophil–monocyte ratio, PDW: platelet distribution width, PLR: platelet–lymphocyte ratio, RBC: red blood cell, RDW: red blood cell distribution width, SII: systemic immune-inflammation index, WBC: white blood cell. ^a^ Significant when compared with the baseline value. ^b^ Significant when compared with the 3rd-month value.

**Table 3 jcm-12-05452-t003:** Comparison of clinical parameters according to the biologics used in the treatment.

		Pre-Treatment	After Treatment (3rd Month)	After Treatment (6th Month)
RBC (10^6^/uL)	Metotrexate	4.78 ± 0.38 4.75 (3.7–6.2)	4.57 ± 0.34 ^a^ 4.68 (3.8–5.6)	4.69 ± 0.34 ^a,b^ 4.81 (3.9–5.8)
	Adalimumab	4.69 ± 0.40 4.75 (4.0–5.6)	4.79 ± 0.59 4.68 (3.9–6.7)	4.76 ± 0.48 4.81 (3.8–6.4)
	Asitretin	4.78 ± 0.40 4.75 (3.8–5.9)	4.80 ± 0.37 4.68 (4.2–6.0)	5.05 ± 1.53 4.81 (4.3–14.7)
	Ustekinumab	4.64 ± 0.35 4.75 (3.8–5.5)	4.68 ± 0.38 4.68 (4.2–5.6)	4.81 ± 0.32 ^a,b^ 4.81 (4.0–5.4)
	Etanercept	4.74 ± 0.34 4.75 (4.1–5.6)	4.43 ± 1.02 ^a^ 4.68 (1.0–5.1)	4.83 ± 0.28 4.81 (4.3–5.6)
	Secukinumab	4.97 ± 0.65 4.75 (4.4–6.6)	4.91 ± 0.71 4.68 (4.1–6.5)	4.87 ± 0.55 4.81 (4.2–6.2)
	Ixekizumab	4.88 ± 0.40 4.77 (4.3–5.5)	4.78 ± 0.44 4.77 (4.2–5.4)	4.63 ± 0.25 4.77 (4.2–4.8)
	Infliximab	4.76 ± 0.26 4.75 (4.3–5.3)	4.55 ± 0.42 ^a^ 4.68 (3.6–5.1)	4.65 ± 0.62 4.81 (3.2–5.4)
HGB (g/dL)	Metotrexate	14.34 ± 1.35 14.26 (10.82–18.87)	14.11 ± 1.23 14.24 (10.6–17.7)	14.13 ± 1.15 ^a^ 14.20 (10.5–17.2)
	Adalimumab	14.01 ± 1.29 14.22 (11.21–16.90)	14.33 ± 1.57 14.24 (11.1–19.87)	14.14 ± 1.18 14.20 (11.4–16.7)
	Asitretin	14.38 ± 1.56 14.35 (9.0–16.52)	14.55 ± 1.09 14.24 (12.2–17.2)	14.37 ± 0.84 14.20 (12.4–17.2)
	Ustekinumab	14.23 ± 1.04 14.26 (11.54–16.60)	14.32 ± 1.53 14.24 (10.7–17.2)	14.46 ± 1.32 14.20 (10.6–16.7)
	Etanercept	14.19 ± 1.18 14.26 (11.85–16.58)	13.99 ± 1.28 14.24 (11.9–16.4)	14.04 ± 0.81 14.20 (11.9–15.3)
	Secukinumab	13.80 ± 1.95 14.10 (10.17–16.68)	13.50 ± 2.12 14.22 (9.8–15.9)	13.74 ± 1.54 14.20 (11.6–15.7)
	Ixekizumab	14.76 ± 1.90 14.59 (11.56–16.75)	14.65 ± 1.72 14.61 (12.6–16.7)	13.81 ± 0.45 13.92 (13.1–14.2)
	Infliximab	14.21 ± 0.59 14.26 (12.90–15.0)	13.75 ± 1.24 ^a^ 14.24 (10.7–14.4)	13.86 ± 1.15 14.20 (11.2–15.1)
HCT (g/dL)	Metotrexate	43.00 ± 3.75 42.67 (32.90–56.50)	42.07 ± 3.45 ^a^ 42.48 (33.1–53.0)	42.78 ± 3.10 43.14 (33.2–50.6)
	Adalimumab	42.00 ± 3.85 41.80 (34.40–51.60)	43.03 ± 5.00 42.48 (34.0–59.8)	42.86 ± 3.48 43.14 (35.2–52.5)
	Asitretin	43.13 ± 4.10 43.40 (30.50–50.60)	43.25 ± 3.39 42.48 (34.7–51.9)	44.54 ± 7.18 43.14 (38–88)
	Ustekinumab	42.29 ± 2.83 42.67 (35.40–48.40)	42.78 ± 4.09 42.48 (31.9–49.8)	43.76 ± 3.57 ^a,b^ 43.14 (33.3–51.4)
	Etanercept	42.52 ± 3.52 42.67 (36.80–50.10)	41.80 ± 3.12 42.48 (35.8–46.4)	41.98 ± 2.90 43.14 (34.9–45.2)
	Secukinumab	41.17 ± 4.96 42.10 (31.60–47.30)	40.51 ± 5.14 42.30 (3.05–46.8)	41.09 ± 3.91 42.5 (35.0–45.6)
	Ixekizumab	43.48 ± 5.01 43.40 (35.70–49.80)	43.44 ± 4.65 43.19 (38.3–49.0)	41.73 ± 1.57 41.97 (39.8–43.1)
	Infliximab	42.50 ± 1.68 42.67 (38.70–44.60)	40.91 ± 3.83 ^a^ 42.48 (31.5–42.5)	42.12 ± 3.66 ^b^ 43.14 (33.2–45.0)
MCV (fL)	Metotrexate	90.82 ± 5.22 90.09 (73–103)	92.17 ± 5.08 ^a^ 91.00 (78.0–103.0)	91.98 ± 4.83 ^a^ 90.45 (78–104)
	Adalimumab	90.40 ± 6.15 90.09 (80–101)	90.72 ± 7.03 90.63 (76–106)	91.50 ± 6.15 90.45 (81–103)
	Asitretin	89.89 ± 4.78 90.09 (73.0–100.0)	89.72 ± 3.33 90.63 (78–98)	88.84 ± 9.72 90.45 (29.9–99.0)
	Ustekinumab	91.42 ± 5.25 90.09 (80.0–106.0)	90.95 ± 5.29 90.63 (73–101)	90.52 ± 5.54 90.45 (73–102)
	Etanercept	89.96 ± 3.68 9.09 (82.0–99.0)	90.52 ± 2.73 90.63 (85–96)	88.59 ± 3.65 ^b^ 90.45 (81.0–93.0)
	Secukinumab	83.34 ± 11.18 87.00 (60–95)	82.55 ± 12.00 75.00 (59–95)	84.87 ± 10.05 90.00 (63.0–93.0)
	Ixekizumab	89.00 ± 8.17 90.50 (74–97)	90.93 ± 6.60 91.50 (79–99)	90.39 ± 3.18 90.45 (85.0–95.0)
	Infliximab	89.68 ± 2.39 90.09 (84–92)	89.89 ± 4.40 90.63 (81–97)	91.91 ± 5.84 90.45 (83.0–103.0)
MCH (pg)	Metotrexate	24.02 ± 12.35 29.92 (0.0–33.7)	30.91 ± 1.76 ^a^ 30.89 (25.8–34.2)	30.49 ± 1.66 ^a,b^ 30.19 (25.2–33.7)
	Adalimumab	24.40 ± 12.34 29.79 (0.0–34.10)	30.24 ± 2.42 30.37 (24.6–34.2)	30.33 ± 2.29 30.19 (26.2–35.2)
	Asitretin	25.74 ± 10.67 30.14 (0.0–32.94)	30.19 ± 1.12 30.37 (26.4–32.6)	30.13 ± 1.33 30.19 (25.7–33.9)
	Ustekinumab	21.72 ± 14.49 30.21 (0.0–34.20)	30.44 ± 2.11 ^a^ 30.37 (22.8–34.8)	30.04 ± 2.14 ^b^ 30.19 (23.2–34.7)
	Etanercept	16.12 ± 15.60 26.40 (0.0–32.86)	30.26 ± 1.48 ^a^ 30.37 (26.9–33.9)	29.83 ± 1.65 ^a^ 30.19 (25.3–31.8)
	Secukinumab	24.59 ± 10.14 28.24 (0.0–33.74)	27.56 ± 4.47 29.04 (19.2–32.4)	28.44 ± 3.88 30.19 (20.3–31.9)
	Ixekizumab	30.27 ± 3.36 30.77 (23.88–33.88)	30.61 ± 2.62 30.76 (25.9–33.7)	30.18 ± 1.42 30.19 (27.8–32.2)
	Infliximab	11.14 ± 15.39 0.00 (0.0–30.60)	30.22 ± 0.87 ^a^ 30.37 (28.3–31.5)	30.50 ± 1.89 ^a^ 30.19 (27.8–34.7)
MCHC (g/dL)	Metotrexate	33.38 ± 0.83 33.45 (31–35.14)	33.55 ± 0.84 33.51 (30.0–35.9)	33.04 ± 0.99 ^b^ 33.04 (30.8–35.6)
	Adalimumab	33.43 ± 1.04 33.45 (30.58–35.16)	33.31 ± 0.87 33.51 (31.0–35.0)	33.04 ± 1.12 33.04 (30.5–35.1)
	Asitretin	33.39 ± 1.31 33.45 (29–35.83)	33.65 ± 0.72 33.51 (32.0–36.1)	32.75 ± 2.70 ^a,b^ 33.04 (30.1–35.2)
	Ustekinumab	33.66 ± 0.84 33.53 (31.86–35.51)	33.47 ± 1.00 33.51 (31.2–36.0)	33.04 ± 0.89 ^a^ 33.04 (30.9–34.8)
	Etanercept	33.39 ± 0.50 33.45 (32.20–34.17)	33.48 ± 0.73 33.51 (31.8–35.3)	33.53 ± 1.05 33.04 (31.0–34.9)
	Secukinumab	33.45 ± 1.03 33.45 (32.22–35.33)	33.24 ± 1.40 32.76 (31.3–35.2)	33.43 ± 0.82 33.27 (32.3–34.5)
	Ixekizumab	33.92 ± 1.32 34.27 (32.35–35.15)	33.68 ± 0.43 33.70 (33.0–34.1)	33.18 ± 0.47 33.04 (32.5–33.9)
	Infliximab	33.40 ± 0.17 33.45 (33.00–33.60)	33.64 ± 0.64 33.51 (32.6–34.9)	33.06 ± 0.50 ^b^ 33.04 (32.0–33.7)
RDW (%)	Metotrexate	13.74 ± 1.32 13.78 (11.4–18.8)	14.52 ± 1.58 ^a^ 14.22 (11.8–20.5)	13.93 ± 1.17 ^b^ 13.83 (11.8–18.0)
	Adalimumab	14.56 ± 1.61 14.15 (11.90–18.30)	14.31 ± 1.53 14.22 (11.9–19.8)	14.09 ± 1.53 13.83 (11.7–18.9)
	Asitretin	13.43 ± 1.21 13.5 (11–17)	14.20 ± 1.97 ^a^ 14.22 (12.5–25.7)	13.60 ± 1.18 13.83 (7.4–16.1)
	Ustekinumab	13.77 ± 1.04 13.87 (11.50–17.10)	13.78 ± 1.50 13.70 (11.8–20.9)	13.76 ± 2.33 13.83 (4.4–20.6)
	Etanercept	13.92 ± 0.86 13.87 (12.40–15.70)	13.93 ± 1.12 14.22 (12.1–16.6)	13.50 ± 1.02 13.83 (11.6–15.9)
	Secukinumab	14.50 ± 1.40 14.90 (11.80–17.00)	14.43 ± 1.88 14.60 (11.9–18.3)	14.14 ± 1.99 13.83 (12.7–19.3)
	Ixekizumab	13.91 ± 2.79 13.30 (11.0–19.3)	14.20 ± 2.90 13.25 (11.8–19.9)	13.85 ± 0.85 13.83 (12.5–15.2)
	Infliximab	14.19 ± 0.51 13.87 (13.87–15.00)	13.87 ± 0.90 14.22 (11.7–14.5)	14.19 ± 1.27 13.83 (12.4–16.7)
WBC (10^3^/uL)	Metotrexate	7.67 ± 1.73 7.73 (4.5–14.7)	7.76 ± 1.68 8.17 (3.9–13.4)	7.44 ± 1.17 7.71 (4.2–10.7)
	Adalimumab	8.14 ± 2.25 7.75 (4.24–13.70)	8.35 ± 20.6 8.17 (5.3–14.3)	7.89 ± 1.63 7.71 (4.0–11.8)
	Asitretin	7.30 ± 2.07 6.87 (3.93–15.31)	7.90 ± 1.39 ^a^ 8.17 (4.8–12.1)	7.59 ± 1.49 7.71 (4.2–12.7)
	Ustekinumab	7.90 ± 1.55 7.25 (4.80–12.60)	8.01 ± 1.80 8.17 (2.2–11.4)	8.09 ± 1.91 7.71 (5.0–12.8)
	Etanercept	7.72 ± 0.92 7.75 (5.40–9.49)	7.68 ± 1.30 8.17 (5.3–10.8)	7.43 ± 1.14 7.71 (6.1–10.7)
	Secukinumab	7.38 ± 1.44 7.75 (5.50–9.50)	13.21 ± 17.04 8.11 (4.3–58.5)	8.09 ± 1.62 8.54 (4.4–9.9)
	Ixekizumab	9.44 ± 1.37 9.37 (7.75–11.55)	8.18 ± 1.24 8.43 (6.3–9.7)	7.67 ± 1.24 7.71 (6.2–9.9)
	Infliximab	7.87 ± 0.67 7.75 (7.10–9.45)	7.67 ± 1.09 8.17 (5.0–8.2)	8.26 ± 0.93 7.71 (7.7–10.0)
Neutrophil(10^3^/uL)	Metotrexate	4.58 ± 1.37 4.64 (2.48–10.80)	4.46 ± 1.23 4.43 (2.2–8.6)	5.01 ± 3.45 4.96 (2.4–31.2)
	Adalimumab	5.10 ± 2.01 4.95 (2.0–10.0)	4.58 ± 1.39 4.43 (2.4–8.3)	4.59 ± 1.27 4.84 (2.4–7.8)
	Asitretin	4.32 ± 1.39 4.25 (2.06–8.54)	4.38 ± 1.03 4.43 (0.9–8.5)	4.66 ± 1.34 4.96 (2.1–10.9)
	Ustekinumab	6.14 ± 8.86 4.95 (2.54–60.40)	4.54 ± 1.53 ^a^ 4.43 (0.9–8.5)	5.57 ± 4.41 ^b^ 4.96 (1.8–30.9)
	Etanercept	4.70 ± 0.86 4.95 (2.80–6.40)	4.30 ± 1.10 4.43 (2.5–7.4)	4.94 ± 1.27 4.96 (3.0–7.8)
	Secukinumab	4.53 ± 1.04 4.95 (2.77–6.05)	4.12 ± 1.48 4.28 (1.8–7.0)	5.08 ± 1.11 ^b^ 4.96 (2.8–6.7)
	Ixekizumab	6.05 ± 1.80 6.06 (4.12–8.90)	4.59 ± 1.21 4.32 (3.5–6.8)	4.97 ± 0.78 4.96 (3.9–6.3)
	Infliximab	5.12 ± 0.77 4.95 (3.94–6.68)	4.22 ± 0.70 ^a^ 4.43 (2.8–5.2)	5.07 ± 0.17 ^b^ 4.96 (5.0–5.5)
Lymphocyte (10^3^/uL)	Metotrexate	2.53 ± 1.50 2.37 (1.22–13.70)	2.33 ± 0.55 2.44 (1.1–3.8)	2.27 ± 0.46 2.42 (0.8–3.4)
	Adalimumab	2.24 ± 0.56 2.34 (0.96–3.27)	2.76 ± 0.69 ^a^ 2.73 (1.4–4.3)	2.54 ± 0.55 ^a^ 2.42 (1.1–3.6)
	Asitretin	2.22 ± 0.68 2.17 (1.35–5.39)	2.42 ± 0.46 ^a^ 2.44 (1.5–4.5)	2.34 ± 0.60 2.42 (0.6–4.4)
	Ustekinumab	2.39 ± 0.51 2.37 (1.15–4.13)	2.48 ± 0.54 2.44 (0.9–4.0)	2.78 ± 1.77 2.42 (1.7–13.4)
	Etanercept	2.39 ± 0.31 2.37 (1.72–3.10)	2.35 ± 0.26 2.44 (1.8–2.8)	2.37 ± 0.25 2.42 (1.9–2.8)
	Secukinumab	2.10 ± 0.61 2.24 (0.99–3.13)	2.13 ± 0.83 2.22 (0.8–3.4)	2.24 ± 0.53 2.42 (1.0–2.7)
	Ixekizumab	2.45 ± 0.81 2.34 (1.56–3.92)	2.61 ± 0.66 2.52 (1.9–3.8)	2.12 ± 0.47 2.42 (1.5–2.5)
	Infliximab	2.14 ± 0.50 2.37 (1.01–2.46)	2.31 ± 0.68 2.44 (1.3–3.5)	2.42 ± 0.35 ^a^ 2.42 (1.9–3.2)
Monocytes (10^3^/uL)	Metotrexate	0.57 ± 0.16 0.57 (0.27–1.32)	0.59 ± 0.19 0.61 (0.2–1.3)	0.57 ± 0.16 0.59 (0.2–1.3)
	Adalimumab	0.58 ± 0.17 0.57 (0.20–1.07)	0.65 ± 0.18 ^a^ 0.61 (0.4–1.3)	0.63 ± 0.17 0.59 (0.3–1.0)
	Asitretin	0.53 ± 0.09 0.53 (0.21–1.12)	0.61 ± 0.11 ^a^ 0.61 (0.3–0.9)	0.56 ± 0.13 ^b^ 0.59 (0.1–1.0)
	Ustekinumab	0.59 ± 0.14 0.57 (0.30–0.95)	0.61 ± 0.16 0.61 (0.2–1.1)	0.59 ± 0.16 0.59 (0.3–1.1)
	Etanercept	0.57 ± 0.09 0.57 (0.40–0.82)	0.60 ± 0.09 0.61 (0.4–0.7)	0.57 ± 0.08 0.59 (0.4–0.8)
	Secukinumab	0.54 ± 0.18 0.48 (0.39–0.98)	0.51 ± 0.11 0.57 (0.4–0.7)	0.57 ± 0.11 0.57 (0.4–0.8)
	Ixekizumab	0.69 ± 0.08 0.69 (0.6–0.84)	0.61 ± 0.09 0.61 (0.5–0.8)	0.55 ± 0.08 0.59 (0.4–0.6)
	Infliximab	0.60 ± 0.07 0.57 (0.57–0.80)	0.61 ± 0.01 0.61 (0.6–0.6)	0.80 ± 0.55 0.59 (0.6–2.2)
Eosinophil (10^3^/uL)	Metotrexate	0.21 ± 0.10 0.20 (0.06–0.71)	0.19 ± 0.07 0.20 (0.0–0.5)	0.19 ± 0.06 0.20 (0.1–0.4)
	Adalimumab	0.19 ± 0.10 0.19 (0.04–0.47)	0.20 ± 0.09 0.20 (0.0–0.5)	0.20 ± 0.11 0.20 (0.0–0.5)
	Asitretin	0.22 ± 0.16 0.20 (0.06–0.85)	0.21 ± 0.09 0.20 (0.1–0.7)	0.21 ± 0.08 0.20 (0.0–0.5)
	Ustekinumab	0.19 ± 0.08 0.20 (0.07–0.58)	0.19 ± 0.08 0.20 (0.1–0.6)	0.20 ± 0.08 0.020 (0.1–0.5)
	Etanercept	0.17 ± 0.04 0.20 (0.09–0.21)	0.18 ± 0.04 0.20 (0.1–0.2)	0.19 ± 0.05 0.20 (0.1–0.3)
	Secukinumab	0.18 ± 0.07 0.20 (0.09–0.35)	0.22 ± 0.17 0.17 (0.0–0.6)	0.27 ± 0.20 0.20 (0.1–0.8)
	Ixekizumab	0.20 ± 0.05 0.18 (0.17–0.32)	0.23 ± 0.07 0.23 (0.1–0.4)	0.21 ± 0.09 0.20 (0.1–0.4)
	Infliximab	0.21 ± 0.06 0.20 (0.11–0.32)	0.19 ± 0.05 ^a^ 0.20 (0.01–0.03)	0.23 ± 0.07 ^b^ 0.20 (0.2–0.4)
Basophil (10^3^/uL)	Metotrexate	0.03 ± 0.01 0.03 (0.01–0.08)	0.03 ± 0.01 0.03 (0.0–0.1)	0.03 ± 0.01 0.03 (0.0–0.1)
	Adalimumab	0.03 ± 0.02 0.03 (0.0–0.09)	0.03 ± 0.01 0.3 (0.0–0.1)	0.03 ± 0.01 0.03 (0.0–0.1)
	Asitretin	0.03 ± 0.01 0.03 (0.0–0.08)	0.03 ± 0.01 0.03 (0.0–0.1)	0.03 ± 0.01 0.03 (0.0–0.1)
	Ustekinumab	0.3 ± 0.1 0.3 (0.0–0.07)	0.03 ± 0.01 0.03 (0.0–0.1)	0.03 ± 0.02 0.03 (0.0–0.1)
	Etanercept	0.03 ± 0.01 0.03 (0.01–0.08)	0.02 ± 0.01 0.03 (0.0–0.1)	0.03 ± 0.01 0.03 (0.0–0.1)
	Secukinumab	0.02 ± 0.01 0.02 (0.01–0.07)	0.3 ± 0.1 0.3 (0.0–0.1)	0.03 ± 0.08 0.03 (0.0–0.1)
	Ixekizumab	0.04 ± 0.01 0.04 (0.02–0.07)	0.03 ± 0.01 0.03 (0.0–0.1)	0.04 ± 0.03 0.03 (0.0–0.1)
	Infliximab	0.02 ± 0.01 0.03 (0.01–0.04)	0.02 ± 0.01 0.03 (0.0–0.0)	0.03 ± 0.01 ^a,b^ 0.03 (0.1–0.3)
Platelets (10^3^/uL)	Metotrexate	269.64 ± 53.94 265.80 (151–408)	254.56 ± 53.22 ^a^ 254.86 (96–401)	253.85 ± 48.15 254.07 (101–386)
	Adalimumab	277.38 ± 61.40 270.50 (140–397)	266.19 ± 73.69 257.43 (71–442)	260.52 ± 49.43 254.07 (148–372)
	Asitretin	249.92 ± 61.37 251 (136–404)	239.37 ± 35.61 254.86 (152–353)	239–84 ± 47.61 254.07 (7.6–352.0)
	Ustekinumab	265.84 ± 47.30 265.80 (114–373)	254.86 ± 62.71 254.86 (104–427)	262.19 ± 59.68 254.07 (133–417)
	Etanercept	279.52 ± 45.63 265.80 (242–389)	263.62 ± 48.67 ^a^ 254.86 (193–355)	270.03 ± 49.45 254.07 (209–386)
	Secukinumab	283.75 ± 64.40 267.0 (186–396)	270.22 ± 82.04 255.00 (148–405)	250.12 ± 78.67 ^a^ 254.07 (130–365)
	Ixekizumab	255.83 ± 74.10 242.50 (162–365)	247.81 ± 80.86 245.43 (158–346)	253.70 ± 39.20 254.07 (186–308)
	Infliximab	263.25 ± 10.03 265.80 (246–279)	251.78 ± 40.87 254.86 (195–334)	252.17 ± 24.16 ^a^ 254.07 (199–286)
MPV (fL)	Metotrexate	8.75 ± 0.62 8.79 (7.4–10.5)	8.78 ± 0.65 8.93 (6.9–10.4)	8.86 ± 0.51 8.89 (7.6–10.1)
	Adalimumab	8.76 ± 0.75 8.79 (6.9–10.2)	8.89 ± 0.63 8.93 (7.7–10.1)	8.80 ± 0.75 8.89 (6.4–10.0)
	Asitretin	8.88 ± 0.75 8.79 (7.3–10.8)	9.05 ± 0.64 8.93 (8.2–11.5)	8.97 ± 0.78 8.89 (7.6–11.8)
	Ustekinumab	8.70 ± 0.71 8.79 (7.2–11.3)	8.74 ± 0.66 8.75 (7.3–10.1)	8.91 ± 0.66 8.89 (7.4–10.5)
	Etanercept	8.87 ± 0.47 8.79 (8.2–10.1)	8.93 ± 0.36 8.93 (8.2–9.9)	8.83 ± 0.43 8.89 (7.7–9.6)
	Secukinumab	8.73 ± 0.87 8.79 (7.6–10.1)	9.36 ± 2.15 8.40 (7.4–14.5)	8.87 ± 1.06 8.70 (7.6–10.6)
	Ixekizumab	8.75 ± 0.49 8.90 (7.8–9.1)	9.13 ± 0.54 8.96 (8.6–10.0)	8.62 ± 0.34 8.79 (8.1–8.9)
	Infliximab	8.78 ± 0.63 8.79 (7.4–9.6)	8.80 ± 0.67 8.93 (7.2–9.5)	8.94 ± 0.33 8.89 (8.4–9.6)
PDW (%)	Metotrexate	14.72 ± 2.05 14.86 (6.40–19.30)	15.14 ± 1.86 15.23 (9.5–21.0)	15.00 ± 1.64 15.00 (11.0–21.5)
	Adalimumab	14.96 ± 1.94 14.86 (9.8–19.3)	15.17 ± 1.86 15.23 (9.8–18.3)	14.90 ± 2.06 15.00 (9.3–18.5)
	Asitretin	15.15 ± 2.25 14.86 (10.5–20.0)	15.76 ± 1.96 15.23 (12.8–22.3)	15.19 ± 1.51 15.00 (12.0–21.0)
	Ustekinumab	14.72 ± 1.95 14.86 (11.30–22.50)	14.93 ± 1.85 15.11 (11.5–20.5)	15.11 ± 2.01 15.00 (11.8–20.8)
	Etanercept	15.00 ± 0.98 14.86 (13.0–16.80)	15.20 ± 1.24 15.23 (12.8–17.5)	14.71 ± 1.38 15.00 (11.8–17.3)
	Secukinumab	13.19 ± 5.39 14.0 (0.0–18.0)	13.70 ± 4.86 13.80 (2.4–20.3)	14.02 ± 5.78 15.00 (0.0–19.8)
	Ixekizumab	15.35 ± 2.12 15.25 (12.0–18.50)	15.77 ± 2.43 15.51 (13.0–19.3)	14.68 ± 1.55 15.00 (12.3–17.0)
	Infliximab	14.98 ± 1.42 14.86 (12.30–17.50)	15.24 ± 1.69 15.23 (11.8–18.0)	14.91 ± 1.04 15.00 (13.5–17.0)
CRP (mg/L)	Metotrexate	5.27 ± 3.20 5.34 (0.8–26.7)	5.10 ± 3.54 ^a^ 4.96 (0.7–29.2)	5.42 ± 5.08 ^b^ 4.87 (0.9–44.8)
	Adalimumab	9.55 ± 15.78 5.34 (1.0–75.0)	5.04 ± 5.54 4.00 (0.4–21.9)	4.36 ± 4.01 ^a^ 3.61 (4.0–22.0)
	Asitretin	5.15 ± 0.86 5.34 (0.7–5.9)	4.96 ± 0.00 ^a^ 4.96 (5.0–5.0)	4.68 ± 0.87 ^a,b^ 4.87 (0.7–4.9)
	Ustekinumab	4.23 ± 4.23 2.85 (0.4–23.0)	4.83 ± 5.15 4.22 (0.7–29.7)	4.07 ± 3.73 4.27 (0.6–20.6)
	Etanercept	3.52 ± 2.60 3.40 (1.0–8.0)	3.51 ± 2.73 2.60 (0.8–11.0)	6.65 ± 12.77 1.80 (0.4–47.7)
	Secukinumab	4.95 ± 1.34 5.34 (2.1–5.3)	7.93 ± 7.56 4.96 (1.4–25.9)	4.52 ± 2.71 4.87 (1.1–10.0)
	Ixekizumab	3.77 ± 1.39 3.65 (2.1–5.3)	3.09 ± 1.66 3.09 (1.2–5.0)	5.00 ± 3.55 4.87 (1.6–11.6)
	Infliximab	4.33 ± 3.55 3.20 (1.1–10.4)	5.38 ± 6.73 3.90 (1.2–21.7)	4.90 ± 2.75 4.87 (1.6–10.3)
ESR (mm/hour)	Metotrexate	16.63 ± 5.84 17.51 (1.0–34.0)	13.93 ± 5.99 ^a^ 13.27 (3.0–45.0)	13.51 ± 2.46 ^a,b^ 13.50 (4.0–25.0)
	Adalimumab	20.58 ± 13.41 17.51 (2.0–67.0)	13.44 ± 9.11 ^a^ 13.27 (1.0–33.0)	15.23 ± 11.39 13.50 (2.0–51.0)
	Asitretin	19.46 ± 10.63 17.51 (12–83)	13.27 ± 0.00 ^a^ 13.27 (13.3–13.3)	12.79 ± 3.09 ^a,b^ 13.50 (1.0–20.0)
	Ustekinumab	17.11 ± 19.58 14.75 (1.0–107.0)	14.49 ± 12.40 13.27 (2.0–60.0)	12.56 ± 8.07 13.50 (1.0–39.0)
	Etanercept	10.91 ± 5.86 11.00 (1.0–19.0)	8.72 ± 4.04 10.0 (1.0–13.3)	10.82 ± 7.93 9.70 (3.0–34.0)
	Secukinumab	23.00 ± 18.12 19.0 (5.0–59.0)	13.75 ± 12.91 13.27 (2.0–6.0)	13.72 ± 13.44 11.00 (3.0–48.0)
	Ixekizumab	12.33 ± 5.74 13.00 (2.0–17.5)	9.63 ± 5.16 12.63 (3.0–13.3)	18.00 ± 10.30 ^b^ 13.50 (1.5–39.0)
	Infliximab	17.75 ± 14.41 16.25 (2.0–49.0)	12.81 ± 8.03 13.27 (2.0–24.0)	18.31 ± 12.56 13.50 (6.0–45.0)
NLR	Metotrexate	1.98 ± 0.63 2.05 (0.44–3.86)	2.01 ± 0.61 1.91 (0.72–4.55)	2.40 ± 1.69 ^a,b^ 2.26 (0.87–14.25)
	Adalimumab	2.49 ± 1.44 2.24 (0.79–8.19)	1.72 ± 0.43 ^a^ 1.88 (0.78–2.51)	1.93 ± 0.58 ^a^ 2.11 (0.78–3.00)
	Asitretin	2.01 ± 0.62 1.95 (1.1–4.33)	1.89 ± 0.46 1.91 (0.94–3.47)	2.34 ± 1.81 ^b^ 2.26 (0.97–13.63)
	Ustekinumab	2.48 ± 2.48 2.21 (1.28–14.46)	1.91 ± 0.74 1.91 (0.97–4.27)	2.19 ± 1.34 2.26 (0.44–9.14)
	Etanercept	2.05 ± 0.49 2.21 (1.35–3.02)	1.87 ± 0.42 1.91 (1.19–2.87)	2.04 ± 0.78 2.26 (1.26–4.22)
	Secukinumab	2.35 ± 0.99 2.00 (1.44–4.29)	2.09 ± 0.78 2.17 (1.04–3.83)	2.37 ± 0.41 2.26 (1.75–2.91)
	Ixekizumab	2.84 ± 1.60 2.82 (1.05–5.71)	1.88 ± 0.76 1.89 (1.12–3.26)	2.50 ± 0.34 2.40 (2.26–3.14)
	Infliximab	2.73 ± 1.27 2.21 (1.60–5.54)	2.03 ± 0.68 ^a^ 1.91 (1.02–3.51)	2.25 ± 0.26 2.26 (1.72–2.68)
NMR	Metotrexate	8.36 ± 2.58 8.46 (2.98–19.85)	8.05 ± 2.21 7.56 (2.76–14.93)	2.27 ± 0.46 ^a,b^ 2.42 (0.84–3.37)
	Adalimumab	9.64 ± 6.56 8.83 (4.27–39.30)	7.17 ± 1.94 ^a^ 7.40 (4.28–13.08)	2.54 ± 0.55 ^a,b^ 2.42 (1.12–3.64)
	Asitretin	8.43 ± 2.17 8.48 (3.96–14.71)	7.37 ± 1.68 ^a^ 7.56 (4.22–13.41)	2.34 ± 0.60 ^a,b^ 2.42 (0.59–4.38)
	Ustekinumab	9.72 ± 9.04 8.78 (5.27–64.26)	7.45 ± 1.71 7.56 (3.19–11.79)	2.79 ± 1.78 ^a,b^ 2.42 (1.66–13.40)
	Etanercept	8.39 ± 1.80 8.83 (5.32–11.43)	7.31 ± 1.45 ^a^ 7.56 (5.18–10.01)	2.37 ± 0.25 ^a,b^ 2.42 (1.85–2.80)
	Secukinumab	8.90 ± 2.96 8.94 (4.01–12.60)	8.03 ± 2.67 7.74 (4.29–2.67)	2.24 ± 0.53 ^a,b^ 2.42 (1.03–2.70)
	Ixekizumab	8.55 ± 1.59 8.71 (6.70–10.60)	7.68 ± 2.73 7.56 (4.55–8.13)	2.42 ± 0.47 ^a,b^ 2.42 (1.52–2.45)
	Infliximab	8.58 ± 1.17 8.83 (6.91–10.60)	7.03 ± 1.22 ^a^ 7.56 (4.55–8.13)	2.42 ± 0.35 ^a,b^ 2.42 (1.92–3.17)
PLR	Metotrexate	118.46 ± 31.01 121.78 (23.43–194.17)	116.16 ± 34.47 111.60 (52.75–219.64)	120.87 ± 35.23 114.57 (66.38–253.10)
	Adalimumab	132.89 ± 41.45 127.02 (47.14–262.10)	100.67 ± 28.96 ^a^ 97.91 (46.88–176.53)	110.79 ± 31.16 ^a^ 114.57 (40.66–183.93)
	Asitretin	118.82 ± 32.11 121.78 (61.01–210.37)	104.15 ± 17.74 ^a^ 111.60 (64.04–149.40)	111.82 ± 35.79 ^b^ 114.57 (12.88–249.12)
	Ustekinumab	117.70 ± 28.81 121.78 (60.29–206.08)	110.15 ± 29.14 111.60 (52.24–192.96)	107.16 ± 35.79 ^a^ 114.50 (18.66–202.05)
	Etanercept	124.39 ± 34.85 121.78 (81.94–226.16)	117.74 ± 30.52 ^a^ 111.60 (83.39–181.46)	119.90 ± 28.34 114.57 (87.52–208.65)
	Secukinumab	145.11 ± 47.99 121.78 (95.65–245.96)	146.96 ± 80.17 111.35 (80.0–323.46)	115.98 ± 28.02 115.02 (50.0–152.72)
	Ixekizumab	112.71 ± 43.84 115.13 (61.83–163.87)	101.07 ± 42.50 97.68 (54.30–166.35)	129.54 ± 36.07 114.57 (108.57–202.63)
	Infliximab	140.44 ± 55.60 121.78 (102.44–276.24)	126.03 ± 57.49 ^a^ 111.60 (59.60–258.91)	111.29 ± 13.77 ^a^ 114.57 (82.65–131.19)
SII	Metotrexate	519.05 ± 226.73 463.40 (140.12–1257.43)	534.24 ± 286.90 464.17 (157.15–1644.34)	617.37 ± 471.16 503.24 (169.94–2963.29)
	Adalimumab	708.39 ± 375.70 624.17 (94.28–1588.00)	451.88 ± 21.390 ^a^ 401.79 (116.25–909.96)	486.55 ± 220.39 ^a^ 445.10 (115.47–1040.63)
	Asitretin	495.31 ± 221.02 467.57 (162.80–1228.56)	414.34 ± 154.92 378.90 (206.70–790.31)	495.28 ± 458.59 396.39 (26.92–2329.88)
	Ustekinumab	693.68 ± 859.26 542.42 (166.41–4992.60)	527.02 ± 338.68 417.20 (111.26–1638.27)	627.52 ± 611.77 437–25 (110.63–3316.60)
	Etanercept	590.98 ± 340.29 445.81 (333.79–1173.78)	523.75 ± 276.16 403.54 (262.31–996.62)	562.72 ± 474.24 402.08 (262.57–1627.46)
	Secukinumab	667.87 ± 349.36 548.35 (454.92–1488.07)	585.35 ± 389.16 449.00 (312.00–1549.18)	594.19 ± 244.07 629.69 (290.00–1015.59)
	Ixekizumab	719.24 ± 388.04 779.01 (265.26–1198.08)	505.14 ± 389.46 301.57 (187.86–1127.83)	709.80 ± 248.27 687.25 (473.57–968.58)
	Infliximab	952.81 ± 572.98 907.88 (403.61–1546.93)	543.68 ± 284.87 685.13 (215.77–730.14)	553.65 ± 114.28 532.73 (451.27–676.95)

CRP: C-reactive protein, ESR: erythrocyte sedimentation rate, HGB: hemoglobin, HTC: hematocrit, MCH: mean erythrocyte hemoglobin, MCHC: mean corpuscular hemoglobin concentration, MCV: mean erythrocyte volume, MPV: mean platelet volume, NLR: neutrophil–lymphocyte ratio, NMR: neutrophil–monocyte ratio, PDW: platelet distribution width, PLR: platelet–lymphocyte ratio, RBC: red blood cell, RDW: red blood cell distribution width, SII: systemic immune-inflammation index, WBC: white blood cell. ^a^ Significant when compared with the baseline value. ^b^ Significant when compared with the 3rd-month value.

**Table 4 jcm-12-05452-t004:** Comparison of the changes in NLR, NMR, and PLR after the treatment.

	Metotrexate	Adalimumab	Asitretin	Ustekinumab	Etanercept	Secukinumab	Ixekizumab	Infliximab
NLR-NLR3	−0.02 ± 0.76 0.11 (−3.16–1.59)	0.77 ± 1.28 0.34 (−0.51–5.68)	0.12 ± 0.58 0.03 (−0.99–2.41)	0.56 ± 2.56 0.30 (−2.04–15.56)	0.17 ± 0.46 0.30 (−0.74–1.13)	0.26 ± 0.89 0.003 (−0.74–1.94)	0.95 ± 1.52 0.68 (−0.48–3.82)	0.70 ± 1.07 0.30 (0.18–3.36)
NLR-NLR6	−0.41 ± 1.82 −0.20 (−12.73–1.96)	0.56 ± 1.44 0.13 (−0.86–5.93)	−0.33 ± 1.95 −0.08 (−12.06–2.06)	0.28 ± 2.82 −0.45 (−7.57–15.20)	0.008 ± 0.73 −0.04 (−2.0–1.28)	−0.18 ± 0.88 −0.26 (−0.96–1.59)	0.33 ± 1.64 −0.03 (−1.21–3.44)	0.48 ± 1.15 −0.45 (−0.12–3.18)
NMR-NMR3	0.30 ± 3.04 0.28 (−7.0–14.34)	2.46 ± 6.57 1.49 (−3.17–32.22)	1.05 ± 2.40 1.02 (−4.58–7.15)	2.26 ± 9.29 0.74 (−5.44–56.71)	1.07 ± 0.88 1.26 (−0.66–3.15)	0.87 ± 2.20 0.93 (−2.64–3.57)	0.86 ± 2.45 1.35 (−3.43–3.80)	1.55 ± 0.56 1.26 (1.17–2.48)
NMR-NMR6	5.96 ± 2.96 6.16 (−6.02–17.72)	7.71 ± 6.46 6.57 (2.93–37.04)	6.08 ± 2.78 6.38 (−5.29–12.45)	7.52 ± 9.21 6.19 (−1.96–61.99)	6.35 ± 1.81 6.57 (4.07–10.04)	6.53 ± 2.82 6.74 (1.75–10.35)	6.04 ± 1.51 5.84 (4.44–8.33)	6.32 ± 1.00 6.57 (4.63–7.93)
PLR-PLR3	2.32 ± 37.91 6.47 (−196.21–72.71)	32.21 ± 42.35 23.02 (−54.74–150.49)	14.66 ± 29.83 10.18 (−43.91–103.08)	7.55 ± 31.11 10.18 (−53.20–93.97)	6.65 ± 25.89 10.18 (−67.45–47.77)	−1.85 ± 66.60 19.17 (−158.6–55.74)	11.63 ± 26.84 6.16 (−15.96–52.85)	14.40 ± 12.01 10.18 (4.15–42.83)
PLR-PLR6	−2.37 ± 42.37 4.20 (−229.67–62.30)	22.10 ± 34.09 7.51 (−33.86–90.96)	6.99 ± 42.50 4.56 (−99.49–108.91)	10.54 ± 31.16 7.21 (−54.29–93.37)	4.48 ± 42.26 7.21 (−86.86–111.59)	29.12 ± 35.43 14.72 (−13.16–93.24)	−16.83 ± 44.61 −36.84 (−52.74–55.30)	29.14 ± 47.71 7.21 (7.21–145.04)

NLR: neutrophil–lymphocyte ratio, NMR: neutrophil–monocyte ratio, PLR: platelet–lymphocyte ratio.

**Table 5 jcm-12-05452-t005:** Correlations of hematological values with each other.

	PASI		CRP		Time Post-Onset		PLR		NMR		NLR	
	*rho*	*p*	*rho*	*p*	*rho*	*p*	*rho*	*p*	*rho*	*p*		
SII	0.201	0.033	0.303	0.013	0.079	0.361	0.612	<0.001	0.477	<0.001	0.848	<0.001
NLR	0.201	0.032	0.263	0.029	0.133	0.124	0.461	<0.001	0.500	<0.001		
NMR	0.118	0.210	0.22	0.066	0.010	0.910	0.138	0.77				
PLR	0.129	0.170	0.031	0.803	0.096	0.265						
Time post-onset	0.153	0.87	−0.089	0.413								
CRP	0.69	0.578										

CRP: C-reactive protein, NLR: neutrophil–lymphocyte ratio, NMR: neutrophil–monocyte ratio, PASI: *Psoriasis Area Severity Index*, PLR: platelet–lymphocyte ratio, SII: systemic immune-inflammation index. SII (Systemic Immune-Inflammation Index) Systemic Immune-Inflammation Index (SII) showed significant positive correlations with Psoriasis Area Severity Index (PASI), C-reactive protein (CRP), Platelet–Lymphocyte Ratio (PLR), and Neutrophil–Lymphocyte Ratio (NLR), indicating potential links between immune and inflammation markers in psoriasis. However, Neutrophil–Monocyte Ratio (NMR) exhibited no significant correlations with other variables, and Time Post-Onset showed no significant correlations with any of the studied markers.

## Data Availability

The data used in this study can be requested from the corresponding author.

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
