# Peer review of "Neutrophil-to-Lymphocyte Ratio, Neutrophil-to-Monocyte Ratio, Platelet-to-Lymphocyte Ratio, and Systemic Immune-Inflammation Index in Psoriasis Patients: Response to Treatment with Biological Drugs"

_jcm, 2023, doi:10.3390/jcm12175452_

Round 1

Reviewer 1 Report

The article touches on an interesting topic and is part of the search for new, simple indicators enabling the advancement of the disease or progress in treatment.

In my opinion, the following corrections are needed:

1. The fragment materials and methods should be supplemented with the conditions in which the material was collected, e.g. was it always in the morning? It is also worth adding what analyzer was used for CBC determinations.

2. The % symbol should be placed after the number

3. Although it is explained in the text, the table column headers should explain what all the values mean.

4. Please complete the units of the tested parameters.

5. I suggest transferring some of the results given in the form of tables to the supplementary file, and enriching the main text with charts.

6. In the discussion, it is worth referring more to the author's own results in the context of other known studies.

7. It is worth discussing the limitations of the study in more detail.

8. The author's affiliation appears to be incomplete. In addition, the plural form should not be used in the work of one author.

9. Some template-required parts of the manuscript are missing - final part.

Author Response

We thank the reviewers and editor for their thoughtful comments.  Our point-by-point response follows, including additional explanations.

Reviewer 2 Report

The research aim is to analyze the evolution of systemic inflammatory markers in psoriasis patients treated with biologic agents.

The abstract is structured appropriately.

The introduction transposes the research into the topic and formulates the objective of the study at the end. However, specific examples on how NLE, PLR, MLR and SII were used to asses systemic inflammation in various medical fields should be provided, for e.g. Moldovan, F.; Ivanescu, A.D.; Fodor, P.; Moldovan, L.; Bataga, T. Correlation between Inflammatory Systemic Biomarkers and Surgical Trauma in Elderly Patients with Hip Fractures. J. Clin. Med. 2023, 12, 5147. doi: 10.3390/jcm12155147.

In the methodology section, the stages of the research are presented. However, it should be divided is subsections (e.g. data collection, statistical analysis, methods of calculating the inflammatory biomarkers etc.). A flow diagram of the studied population could be provided for a better overview.

The results are clearly described indicating the measured values. No reference is made in text to Table 3.  

The discussions interpret the research results and relate them to other scientific papers. Limitations of the study should be provided at the end of the discussion section in a new paragraph.

The conclusion should be stated in a different section.

The references are adequate but should be extended as suggested above.

Editing suggestion - the sections should be numbered.

Author Response

(The authors gave the same response as above.)

Round 2

Reviewer 1 Report

The authors have addressed my remarks in a thorough and satisfactory fashion.